# Vitamin D deficiency, impaired lung function and total and respiratory mortality in a cohort of older men: cross-sectional and prospective findings from The British Regional Heart Study

S Goya Wannamethee ,[1] Paul Welsh,[2] Olia Papacosta,[3] Lucy Lennon,[3] Peter Whincup[4]

¹UCL Department of Primary Care & Population Health, UCL Medical School, London, UK
²Institute of Cardiovascular and Medical Sciences, University of Glasgow, Glasgow, UK
³Primary Care and Population Health, UCL Medical School, London, UK
⁴Population Health Research Institute, St George's University of London, London, UK

**Correspondence to**
S Goya Wannamethee;
g.wannamethee@ucl.ac.uk

## ABSTRACT

**Objectives** Vitamin D deficiency is associated with chronic obstructive pulmonary disease (COPD). We examined the cross-sectional association between 25-hydroxyvitamin D (25(OH)D) and lung function impairment and assessed whether vitamin D deficiency is related to long-term mortality in those with impaired lung function.

**Design** Prospective study

**Setting** General practices in the UK.

**Participants** 3575 men aged 60–79 years with no prevalent heart failure.

**Outcome measures** Airway obstruction and mortality. The Global Initiative on Obstructive Lu
ng diseases (GOLD) spirometry criteria was used to define airway obstruction.

**Results** During the follow-up period of 20 years, there were 2327 deaths (114 COPD deaths). Vitamin D deficiency was defined as serum 25(OH)D levels <10 ng/mL; insufficiency as 25(OH)D 10–19 ng/mL; sufficient as 25(OH)D >20 ng/mL. In cross-sectional analysis, vitamin D deficiency was more prevalent in those with moderate COPD (FEV/FVC <70% and $FEV_1$ 50 to <80%; $FEV_1$, forced expiratory volume in 1 s and FVC, forced vital capacity) and severe COPD (FEV/FVC <70% and $FEV_1$ <50%) but not in those with mild COPD (FEV/FVC <70% and $FEV_1$ ≥80%) or restrictive lung disease ($FEV_1$/FVC ≥70% and FVC <80%) compared with men with normal lung function. Vitamin D deficiency was associated with increased risk of total and respiratory mortality in both men with COPD and men with restrictive lung disease after adjustment for confounders and inflammation. The adjusted HRs (95% CI) for total mortality comparing levels of 25(OH)D <10 ng/mL to 25(OH)D >=20 ng/mL were 1.39 (1.10 to 1.75), 1.52 (1.17 to 1.98), 1.58 (1.17 to 2.14) and 1.39 (0.83 to 2.33) for those with no lung impairment, restrictive lung function, mild/moderate COPD and severe COPD, respectively.

**Conclusion** Men with COPD were more likely to be vitamin D deficient than those with normal lung function. Vitamin D deficiency is associated with increased all-cause mortality in older men with no lung impairment as well as in those with restrictive or obstructive lung impairment.

---

### Strengths and limitations of this study

► This study is carried out in a high-risk group for vitamin D deficiency and chronic obstructive lung disease and examined restrictive and obstructive lung disease separately.

► The study population is socially representative of the UK.

► The study took into account a wide range of confounders and inflammatory markers that could explain the associations between vitamin D deficiency and lung function and mortality.

► The study is limited by the use of a single measurement of vitamin D (25(OH)D).

► The study is restricted to a predominantly white male population of European extraction,

---

## INTRODUCTION

Chronic obstructive lung disease (COPD) is a common chronic inflammatory disease in the elderly and is a major cause of morbidity and mortality.[1] Much attention has focused on identifying factors which may influence prognosis and mortality in these patients. Vitamin D deficiency (ascertained from measuring the circulating 25-hydroxyvitamin D (25(OH)D) metabolite) is recognised as an important health problem particularly in the elderly[2] and has been associated with various diseases including respiratory infections, autoimmune disease, cardiovascular disease (CVD), diabetes and cancer.[3–6] Although vitamin D deficiency is traditionally known for its role in bone health, there is growing interest in the role of vitamin D in the pathogenesis and severity of COPD.[7] Observational studies have found vitamin D deficiency to be highly prevalent in patients with COPD.[8–12] While many cross-sectional population studies have associated higher 25(OH)D with better lung

function as measured by the FEV1,[10–16] prospective studies of the association of baseline 25(OH)D with lung function decline and incident COPD have produced conflicting results.[9–11 15 16] Whether the association between 25(OH)D and impaired lung function is specific to COPD is also unclear. Vitamin D deficiency has been associated with increased mortality in the general population,[17–19] but the associations with pulmonary death and total mortality in COPD patients remains uncertain. Longitudinal studies that have examined the association between 25(OH)D and mortality in COPD patients have mostly reported no significant association with mortality.[20–24] However, most of these previous studies have targeted moderate and severe COPD patients only. Two recent prospective studies showed that vitamin D deficiency or low vitamin D was associated with all-cause mortality in those with COPD in the general population.[25 26] Few studies have examined the association between vitamin D deficiency and impaired lung function in the general older population specifically and whether vitamin D deficiency is associated with mortality in older adults with milder degrees of impaired lung function (restrictive or obstructive) has not been well studied. The aim of the present study is to (1) investigate the cross-sectional association between vitamin D deficiency and both restrictive and obstructive lung function impairment and (2) whether vitamin D deficiency is related to long-term mortality in those with impaired lung function.

## Subjects and methods
The British Regional Heart Study is a prospective study involving 7735 men aged 40–59 years drawn from one general practice in each of 24 British towns, who were screened between 1978 and 1980.[27] The population studied was socioeconomically representative of British men and comprises predominantly white Europeans (>99%). In 1998–2000, all surviving men, then aged 60–79 years, were invited for a 20th year follow-up examination. All men provided informed written consent. All men completed a mailed questionnaire providing information on their lifestyle and medical history, had a physical examination and provided a fasting blood sample. The samples were frozen and stored at −20°C on the day of collection and transferred in batches for storage at −70°C until analysis, carried out after no more than one freeze–thaw cycle. The men were asked whether a doctor had ever told them that they had angina or myocardial infarction, heart failure (HF) or stroke; details of their medications were recorded at the examination including use of bronchodilators (BNF code 3.1). 4252 men (77% of available survivors) attended for examination.

## Study subjects
Blood measurements of 25(OH)D were available in 3754 men (88%) at the 20-year follow-up examination (1998–2000). Of these men, we excluded 88 men with a history of a diagnosis of HF as these men have exceptional mortality rates and reduced lung function and a further 91 men with no lung function measures leaving 3575 men for analysis.

## Cardiovascular risk factor measurements at 1998–2000
Anthropometric measurements including body weight, height and waist circumference were carried out. Details of measurement and classification methods for smoking status, physical activity, social class, alcohol intake, blood pressure and blood lipids in this cohort have been described.[28 29] Heavy drinking was defined as drinking more than five units (1 UK unit=10g) of alcohol daily on most days. Physical activity scores were assigned on the basis of frequency and type of activity and the men were divided into six groups: none, occasional, light, moderate, moderately vigorous and vigorous. Subjects who reported none or occasional activity were classified as 'inactive'. On the basis of their reported cigarette smoking status, the men were classified as never smokers, long-term ex-smokers (≥15 years), recent ex-smokers (<15 years) and current smokers. Social class was derived from the longest held occupation recorded at the time of baseline questionnaire (1978–1980) using the Registrar General's classification of occupations, with categories grouped as non-manual (I, II and III non-manual) and manual (III manual, IV and V). Interleukin-6 was assayed using a high-sensitivity ELISA (R&D Systems, Oxford, UK). C-reactive protein (CRP) was assayed by ultra-sensitive nephelometry (Dade Behring, Milton Keynes, UK).

## Vitamin D
Measurement of 25(OH)D was performed on EDTA-anticoagulated plasma via a high-throughput method for the measurement of $25(OH)D_3$ and $25(OH)D_2$ using a gold-standard liquid chromatography-tandem mass spectrometry method following an automated solid-phase extraction procedure.[30 31] Our method is calibrated and controlled using reagents from Chromsystems GmbH (Manchester, UK) and is currently in routine clinical use. Results are reported as total 25(OH)D ($25OHD_2$+$25OHD_3$); virtually all participants had an undetectable $25OHD_2$, which is commensurate with results observed in routine NHS use. The lower limit of sensitivity was 4ng/mL for both $25(OH)D_3$ and total 25(OH)D. Vitamin D deficiency was defined as 25(OH)D<10ng/mL and vitamin D insufficiency as 25(OH)D 10–19ng/mL.[2 31 32]

## Lung function
Forced expiratory volume in 1s ($FEV_1$) and forced vital capacity (FVC) were measured using a Vitalograph Compact spirometer with the subject seated.[29] We used predictive equations for $FEV_1$ and FVC that were derived from the general population Health Survey for England for males aged >25 years.[33] The following equations were used: predicted $FEV_1$=−exp[−9.37674+0.000183×age−0.00011×$age^2$+log(height)] predicted FVC=exp(−10.36706+0.00434×age−0.00011×$age^2$+log(height)).

**Table 1** Baseline characteristics according to 25(OH)D groups in men without history of heart failure

| | 25(OH)D (ng/mL) | | | |
| --- | --- | --- | --- | --- |
| | <10 (n=363) | 10–19 (n=1499) | ≥20 (n=1713) | *P -value trend* |
| Age (years) | 69.6 (5.8) | 68.7 (5.5) | 68.4 (5.4) | *0.0003* |
| % smokers | 25.3 | 12.8 | 9.5 | *<0.0001* |
| % manual | 56.2 | 52.7 | 54.9 | *0.77* |
| % inactive | 56.4 | 34.9 | 27.9 | *<0.0001* |
| % heavy drinkers | 6.1 | 2.9 | 3.7 | *0.01* |
| % MI/stroke | 18.2 | 13.6 | 13.4 | *0.07* |
| % diabetes | 16.5 | 14.1 | 11.0 | *<0.0001* |
| % antihypertensive | 38.6 | 32.0 | 31.6 | *0.04* |
| % renal dysfunction | 15.4 | 15.6 | 14.9 | *0.63* |
| BMI (kg/m$^2$) | 26.8 (4.5) | 27.1 (3.8) | 26.6 (3.3) | *0.007* |
| % obese | 19.6 | 18.5 | 13.3 | *<0.0001* |
| SBP (mm Hg) | 152.4 (26.1) | 148.6 (23.9) | 149.1 (23.6) | *0.16* |
| CRP (mg/L)* | 2.25 (1.00-5.05) | 1.65 (0.86-3.48) | 1.57 (0.78-3.10) | *<0.0001* |
| IL-6* | 3.0 (1.87-4.43) | 2.50 (1.61-3.59) | 2.27 (1.48-3.09) | *<0.0001* |
| **Lung function** | | | | |
| FEV$_1$ (L) | 2.35 (0.71) | 2.58 (0.65) | 2.67 (0.64) | *<0.0001* |
| % FEV predicted | 80 (23) | 86 (21) | 89 (21) | *<0.0001* |
| Abnormal FEV$_1$ | 12.3 | 4.9 | 3.9 | *<0.0001* |
| FVC (L) | 3.17 (0.90) | 3.36 (0.85) | 3.47 (0.83) | *<0.0001* |
| % FVC predicted | 81 (20) | 85 (19) | 88 (19) | *<0.0001* |
| Abnormal FVC | 9.1 | 5.5 | 4.1 | *<0.0001* |
| FEV$_1$/FVC | 0.74 (0.13) | 0.77 (0.12) | 0.77 (0.11) | *<0.0001* |
| % COPD | 32.5 | 22.0 | 21.0 | *<0.0001* |
| Mild | 6.3 | 6.5 | 7.8 | *0.16* |
| Moderate | 15.4 | 11.5 | 9.6 | *0.001* |
| Severe | 10.7 | 3.9 | 3.6 | *<0.0001* |
| % restrictive lung disorder | 28.7 | 30.2 | 26.5 | *0.07* |

Mean and SD unless specified.
% refers to the % of men with the characteristics within the 25(OH)D groups.
Italics numbers are p value
*Geometric mean and IQR.
BMI, body mass index; COPD, chronic obstructive pulmonary disease; CRP, C-reactive protein; FEV1, forced expiratory volume in 1 s; FVC, forced vital capacity; MI, myocardial infarction; SBP, systolic blood pressure.

Predicted FEV$_1$%=raw FEV$_1$/predicted FEV$_1$ and predicted FVC%=raw FVC/predicted FVC. We divided the men into five lung pattern groups according to their FEV$_1$/FVC ratio using a modification of the criteria developed by The Global Initiative for Chronic Obstructive Lung Disease (GOLD) for defining the severity stages of obstructive airways:[34] (1) severe airflow obstruction (FEV$_1$/FVC <0.70 and FEV$_1$ <50% predicted), moderate airflow obstruction (FEV$_1$/FVC <0.70 and FEV$_1$≥50 to<80%), mild airflow obstruction (FEV$_1$/FVC <0.70 and FEV1 ≥80%), restricted (FEV$_1$/FVC≥0.70 and FVC <80% predicted) and normal (FEV$_1$/FVC≥0.70 and FVC≥80% and not on bronchodilators).

**Follow-up**

All men have been followed up from initial examination (1978–1980) for cardiovascular morbidity[27] and follow-up has been achieved for 99% of the cohort. In the present analyses, all-cause mortality is based on follow-up from re-examination in 1998–2000 at mean age 60–79 years to September 2019. Details on causes of deaths were collected through the National Health Service Central Register. COPD deaths were defined as deaths due to chronic bronchitis (ICD-9 codes 490–491), emphysema (ICD-9 code 492) and chronic airway obstruction (ICD-9 code 496).

**Statistical analysis**

The men were classified into three vitamin D groups: 25(OH)D<10 ng/mL (deficient), 10–19 ng/mL (insufficient) and ≥20 ng/mL (sufficient). Cox's proportional hazards model was used to assess the multivariate-adjusted HR (relative risk) for mortality in a comparison of three vitamin D groups: <10, 10–19 and ≥20 ng/mL and as well as in a one unit increase in 25OHD.

**Table 2** Restrictive and obstructive lung function patterns and cross-sectional associations with circulating 25(OH)D and vitamin D deficiency

| | | | Obstructive | | |
|---|---|---|---|---|---|
| | Normal n=1756 | Restrictive n=1011 | Mild n=254 | Moderate n=394 | Severe n=160 |
| FEV$_1$,% predicted | 101 | 75 | 95 | 66 | 39 |
| FVC,% predicted | 96 | 67 | 101 | 83 | 61 |
| 25OHD (ng/mL) | | | | | |
| Mean (SD) | 20.71 (0.21) | 19.62 (0.29) | 20.95 (0.61) | 18.86 (0.81) | 17.60 (0.74) |
| Adjusted difference in mean 25(OH)D (95% CI) * | | | | | |
| Age and season adjusted | 0 | −1.05 (−1.76 to −0.32) | 0.38 (−0.82 to 1.58) | −1.74 (−2.74 to −0.74) | −2.86 (−3.36 to −1.36) |
| Model 1 | 0 | −0.69 (−1.38 to 0.00) | 1.15 (−0.03 to 2.33) | −0.98 (−1.96 to −0.01) | −1.51 (−2.96 to −0.06) |
| Model 2 | 0 | −0.52 (−1.21 to 0.20) | 1.00 (−0.18 to 2.18) | −0.85 (−1.85 to 0.16) | −1.21 (−2.66 to 0.24) |
| % with vitamin D deficiency | 8.0 | 10.3 | 9.1 | 14.2 | 24.4 |
| Relative odds (95% CI) of having vitamin D deficiency | | | | | |
| Age and season adjusted | 1.00 | 1.36 (1.04 to 1.78) | 0.99 (0.62 to 1.59) | 1.82 (1.30 to 2.55) | 3.52 (2.33 to 5.30) |
| Model 1 | 1.00 | 1.09 (0.82 to 1.45) | 0.85 (0.51 to 1.40) | 1.49 (1.04 to 2.14) | 2.13 (1.35 to 3.35) |
| Model 2 | 1.00 | 1.01 (0.75 to 1.35) | 0.89 (0.54 to 1.48) | 1.47 (1.03 to 2.11) | 1.98 (1.26 to 3.14) |

Model 1: adjusted for age, season, smoking, physical activity, social class, BMI, diabetes and pre-existing CVD.
Model 2=model 1+IL-6.
*Difference in mean 25(OH)D compared with those with normal lung function.
BMI, body mass index; FEV1, forced expiratory volume in 1 s; FVC, forced vital capacity.

All analyses were initially adjusted for age and season. In the multivariate analyses, we adjusted further for potential confounders known to be associated with lung function and mortality which included smoking, social class, physical activity, heavy drinking, BMI use of antihypertensive treatment, diabetes and pre-existing CVD. We also carried out supplementary analysis without inclusion of BMI since BMI may be a confounder or a potential mediator. We further adjusted for IL-6 as a potential mediator. In multivariate analyses, smoking, social class, physical activity, heavy drinking, diabetes, use of antihypertensive treatment and pre-existing CVD were fitted as categorical variables; IL-6 and BMI were fitted as a continuous variable. Analysis of variance was used to obtain adjusted mean differences in 25(OH)D between the lung impairment groups. Logistic regression was used to assess the relative odds of having vitamin D deficiency (yes/no) for the lung impairment groups compared with those with normal lung function.

## Patients and public involvement

Patients and the public were not involved in the conduct, design, analysis and interpretation of this research.

## RESULTS

During the mean follow-up period of 20 years, there were 2327 deaths from all causes in the 3575 men with no diagnosed HF, including 114 deaths due to COPD. The mean 25(OH)D was lower in men examined in winter (December –February) than in those examined in summer (June–August) 17.1 (7.4) versus 22.2 (9.4) ng/mL. The prevalence of COPD was also much higher in winter than in summer (30.2% vs 17%).

## Baseline characteristics

Table 1 summarises baseline characteristics by the three 25(OH)D groups. At baseline, men with vitamin D deficiency had a more adverse health profile including the highest prevalence of smoking, heavy drinking, physical inactivity, obesity, hypertension and the highest mean CRP and IL-6 (markers of inflammation) (table 1). 25.3% of those with vitamin D deficiency were smokers compared with only 9.5% in those who had vitamin D sufficiency. Lung function (both FEV and FVC) was higher in those with higher 25(OH)D levels. Men with vitamin D deficiency had the highest prevalence of COPD and this was seen for moderate and severe COPD but not for mild COPD. There was no association between vitamin D deficiency and prevalence of restrictive lung impairment. We divided the vitamin D sufficient group (25(OH)D>20 ng/mL) further into those with ranges 20–29, 30–39 and those with ≥40 ng/mL. There was no further increase in lung function with increasing levels of 25(OH)D among the sufficient group (online supplemental Table 1).

## Cross sectional associations between restrictive and obstructive lung function patterns and 25(OH)D and vitamin D deficiency

Table 2 shows the absolute mean 25(OH)D for the five lung function groups and the adjusted difference in mean 25(OH)D for those with impaired lung function

**Table 3** Circulating 25(OH)D and adjusted HR (95% CI) for total CVD, respiratory and COPD mortality

| | 25(OH)D (ng/mL) | | | | |
|---|---|---|---|---|---|
| | <10 (deficient) (n=363) | 10–19 (n=1499) | 20+ (n=1713) | HR per 1 ng/mL increase in 25(OH)D | P-value linear trend |
| **Total** | | | | | |
| Rate/1000 | 67.8 (287) | 46.4 (988) | 40.9 (1052) | | |
| Age +season | 1.80 (1.57, 2.05) | 1.16 (1.06, 1.27) | 1.00 | 0.983 (0.978, 0.988) | <0.0001 |
| Model 1 | 1.52 (1.32, 1.75) | 1.14 (1.04, 1.25) | 1.00 | 0.989 (0.984, 0.994) | <0.0001 |
| Model 2 | 1.56 (1.32, 1.75) | 1.14 (1.04, 1.25) | 1.00 | 0.989 (0.984, 0.994) | <0.0001 |
| Model 3 | 1.47 (1.28, 1.70) | 1.12 (1.02, 1.22) | 1.00 | 0.990 (0.985, 0.995) | <0.0001 |
| **CVD** | | | | | |
| Rate/1000 | 20.6 (87) | 15.6 (332) | 14.0 (359) | | |
| Model 1 | 1.27 (0.99, 1.64) | 1.11 (0.95, 1.30) | 1.00 | 0.992 (0.983, 1.000) | 0.06 |
| Model 2 | 1.28 (0.99, 1.65) | 1.11 (0.95, 1.30) | 1.00 | 0.992 (0.983, 1.000) | 0.06 |
| Model 3 | 1.23 (0.95, 1.59) | 1.07 (0.92, 1.26) | 1.00 | 0.993 (0.985, 1.002) | 0.15 |
| **Respiratory** | | | | | |
| Rate/1000 | 11.4 (48) | 4.8 (103) | 4.4 (113) | | |
| Model 1 | 2.10 (1.46, 3.02) | 1.08 (0.82, 1.43) | 1.00 | 0.978 (0.963, 0.993) | 0.004 |
| Model 2 | 2.07 (1.44, 2.99) | 1.08 (0.82, 1.42) | 1.00 | 0.978 (0.964, 0.993) | 0.005 |
| Model 3 | 2.01 (1.39, 2.90) | 1.06 (0.80, 1.40) | 1.00 | 0.979 (0.964, 0.994) | 0.007 |
| **COPD** | | | | | |
| Rate/1000 | 5.7 (24) | 2.3 (48) | 1.6 (42) | | |
| Model 1 | 2.28 (1.32, 3.91) | 1.42 (0.92, 2.17) | 1.00 | 0.972 (0.950, 0.994) | 0.01 |
| Model 2 | 2.25 (1.31, 3.88) | 1.43 (0.93, 2.19) | 1.00 | 0.972 (0.950, 0.994) | 0.01 |
| Model 3 | 2.06 (1.19, 3.58) | 1.37 (0.88, 2.11) | 1.00 | 0.975 (0.953, 0.997) | 0.02 |

Model 2=model 1+BMI.

Model 3=model 2+IL-6.

Model 1: adjusted for age, season, smoking, physical activity, social class, diabetes, use of antihypertensive treatment, pre-existing CVD and impaired lung function.

BMI, body mass index; COPD, chronic obstructive pulmonary disease; CVD, cardiovascular disease;

compared with those with normal lung function. Mean 25(OH)D was lower in men with moderate or severe airway obstruction but not in those with mild airway obstruction compared with those with normal function (table 2). Men with moderate and severe COPD but not those with restrictive lung patterns were more likely to have vitamin D deficiency. Men with moderate and severe COPD showed increased odds of having vitamin D deficiency compared with those with normal lung function even after adjustment for potential confounders including age, season, smoking, physical activity, social class, heavy drinking, BMI, diabetes and pre-existing CVD (table 2). These associations persisted after further adjustment for IL-6.

### Prospective associations between 25(OHD) and total mortality

25(OH)D was inversely associated with total mortality even after adjustment for potential confounders, BMI and inflammation (IL-6) (table 3). The inverse association persisted on exclusion of current smokers. Compared with those with vitamin D sufficiency the adjusted HR (95% CI) were 1.41 (1.19 to 1.68) and 1.11 (1.00 to 1.24) for those with vitamin D deficiency and insufficiency respectively (p=0.002 for trend). The inverse association was seen for mortality from respiratory and COPD causes but not from CVD causes after adjustment for potential confounders, BMI and inflammation (table 3).

When examined by lung function status, vitamin D deficiency was associated with increased mortality in those with normal lung function as well as in those with restrictive lung disorder or mild/moderate COPD even after adjustment for confounders and inflammation (table 4). In men with severe COPD, a statistically non-significant increase in mortality was seen possibly due to small numbers. Vitamin D deficiency was associated with increased risk of respiratory mortality even after adjustment although this was statistically significant only in the restrictive and mild and moderate COPD group possibly because of small numbers (table 5). The number of COPD deaths was small, but vitamin D deficiency was associated with increased risk of COPD

**Table 4** Circulating 25(OH)D and adjusted HR (95% CI) for total mortality by patterns of lung function

| | 25(OH)D (ng/mL) | | | |
| --- | --- | --- | --- | --- |
| | <10 (deficient) | 10–19 | 20+ | *P-linear trend* |
| None (n=1756) | | | | |
| Rate/1000 (n/N) | 52.9 (100/141) | 35.7 (397/716) | 34.4 (490/899) | |
| Model 1 | 1.46 (1.16, 1.84) | 1.07 (0.93, 1.23) | 1.00 | *0.02* |
| Model 2 | 1.45 (1.15, 1.82) | 1.06 (0.92, 1.22) | 1.00 | *0.02* |
| Model 3 | 1.39 (1.10, 1.75) | 1.05 (0.91, 1.22) | 1.00 | *0.04* |
| Restrictive (n=1011) | | | | |
| Rate/1000 (n/N) | 74.4 (86/104) | 51.8 (341/453) | 41.6 (305/454) | |
| Model 1 | 1.54 (1.18, 2.00) | 1.24 (1.05, 1.46) | 1.00 | *0.002* |
| Model 2 | 1.52 (1.19, 2.00) | 1.27 (1.08, 1.50) | 1.00 | *0.002* |
| Model 3 | 1.52 (1.17, 1.98) | 1.26 (1.07, 1.49) | 1.00 | 0.002 |
| Mild/moderate COPD (n=648) | | | | |
| Rate/1000 (n/N) | 75.3 (65/79) | 56.7 (199/271) | 49.0 (207/298) | |
| Model 1 | 1.68 (1.25, 2.26) | 1.18 (0.96, 1.45) | 1.00 | *0.02* |
| Model 2 | 1.68 (1.25, 2.25) | 1.18 (0.96, 1.45) | 1.00 | *0.04* |
| Model 3 | 1.58 (1.17, 2.14) | 1.11 (0.90, 1.36) | 1.00 | *0.04* |
| Severe COPD (n=160) | | | | |
| Rate/1000 (n/N) | 96.4 (36/39) | 71.8 (51/59) | 65.0 (50/62) | |
| Model 1 | 1.41 (0.85, 2.35) | 0.87 (0.57, 1.33) | 1.00 | *0.56* |
| Model 2 | 1.43 (0.85, 2.38) | 0.87 (0.57, 1.34) | 1.00 | *0.56* |
| Model 3 | 1.39 (0.83, 2.33) | 0.85 (0.55, 1.30) | 1.00 | *0.64* |

Model 1: adjusted for age, season, smoking, physical activity, social class, diabetes, use of antihypertensive treatment and pre-existing CVD.
Model 2=model 1+BMI.
Model 3=model 2+IL-6.
BMI, body mass index; COPD, chronic obstructive pulmonary disease; CVD, cardiovascular disease.

mortality in all groups with the exception of those with severe COPD.

### DISCUSSION
Our findings confirm previous observational population studies on the association between low 25(OH)D and COPD and total mortality[13–19] and extend these findings further. The association was not explained by smoking, physical inactivity or inflammation (IL-6) which are known to relate to both COPD and vitamin D deficiency.[35] Restrictive lung function, by contrast, was not associated with vitamin D deficiency. The definition of vitamin D deficiency and insufficiency remains controversial; the UK's National Institute for Health and Care Excellence defines it as <25 nmol/L (<10 ng/mL),[36] while the US' Institute of Medicine defines it as <30 nmol/L (<12.5 ng/mL).[37] We have shown that 25(OH)D<10 ng/mL which is regarded as deficient[32] to be associated with increased risk of total mortality and COPD mortality in those with restrictive and mild/moderate COPD but not in those with severe COPD which was independent of known potential confounders.

25(OH)D levels of 50 nmol/L (20 ng/mL) or more are sufficient for most people.[37] In contrast, the Endocrine Society stated that, for clinical practice, a serum 25(OH)D concentration of more than 75 nmol/L (30 ng/mL) is necessary to maximise the effect of vitamin D on calcium, bone and muscle metabolism.[38] When those with 25(OH)D concentration >20 ng/mL were divided further, there was no evidence that those with levels 20–29 ng/mL had lower lung function or higher mortality than those with levels>30 ng/mL. There has been suggestion that mortality is also increased in those with higher extreme 25(OH)D levels (>100 nmol/L or 40 ng/mL).[39] The number of older men with levels>40 ng/mL in this study was small and we observed no excess mortality in this group in contrast to the Copenhagen Vitamin D Study where the average age (50 years) was younger.[39]

### Vitamin D deficiency and COPD
It is now established that vitamin D deficiency (variously defined) is highly prevalent in patients with COPD.[6–9 12] In line with these studies, we have shown in cross-sectional analysis that those with COPD were more likely to have vitamin D deficiency (<10 ng/mL) than those with normal

**Table 5** Circulating 25(OH)D and adjusted HRs (95% CI) for respiratory and COPD mortality by patterns of lung function

| | 25(OH)D (ng/mL) | | |
| --- | --- | --- | --- |
| | <10 (deficient) | 10–19 | 20+ |
| **Respiratory deaths** | | | |
| None | | | |
| Rates/1000 p-years | 4.2 (8/141) | 2.6 (29/716) | 2.8 (40/899) |
| Adjusted* HR | 1.63 (0.74, 3.62) | 0.99 (0.60, 1.62) | 1.00 |
| Restrictive | | | |
| Rates/1000 p-years | 11.6 (13/104) | 5.8 (35/453) | 4.9 (32/454) |
| Adjusted HR | 2.34 (1.16, 4.71) | 1.26 (0.75, 2.10) | 1.00 |
| Mild/moderate COPD | | | |
| Rates/1000 p-years | 15.0 (13/79) | 6.8 (24/271) | 6.2 (26/298) |
| Adjusted HR | 2.91 (1.42, 5.97) | 1.16 (0.65, 2.08) | 1.00 |
| Severe COPD | | | |
| Rates/1000 p-years | 23.3 (14/39) | 20.5 (15/59) | 20.5 (15/62) |
| Adjusted HR | 1.27 (0.53, 3.03) | 0.65 0.28, 1.50) | 1.00 |
| **COPD deaths** | | | |
| None | | | |
| Rates/1000 p-years | 1.0 (2/141) | 0.4 (4/716) | 0.4 (6/899) |
| Adjusted HR | 4.78 (0.86, 26.70) | 1.13 (0.30, 4.29) | 1.00 |
| Restrictive | | | |
| Rates/1000 p-years | 2.7 (3/104) | 2.8 (17/453) | 0.9 (6/454) |
| Adjusted HR | 3.26 (0.72, 14.80) | 3.64 (1.36, 9.71) | 1.00 |
| Mild/moderate COPD | | | |
| Rates/1000 p-years | 11.4 (9/79) | 4.9 (17/271) | 3.9 (16/298) |
| Adjusted HR | 3.75 (1.55, 9.12) | 1.46 (0.71, 3.00) | 1.00 |
| Severe COPD | | | |
| Rates/1000 p-years | 28.2 (10/39) | 15.5 (10/59) | 19.1 (14/62) |
| Adjusted HR | 1.04 (0.40, 2.73) | 0.47 (0.18, 1.26) | 1.00 |

*Adjusted for age, season, smoking, physical activity, social class, diabetes, use of antihypertensive treatment, pre-existing CVD and BMI.
BMI, body mass index; COPD, chronic obstructive pulmonary disease; CVD, cardiovascular disease.

lung function and the prevalence of vitamin D deficiency increased with increasing severity of COPD even after taking into account a wide range of possible confounders and mediators including smoking, physical inactivity, BMI and inflammation (IL-6). Whether vitamin D deficiency is a consequence of COPD or whether it plays a role in the development of COPD is still unclear.[7] Despite the biological evidence that vitamin D may improve lung function through its action on regulating inflammation, inducing antimicrobial peptides and/ or its action on muscle,[6 35] the potential for reverse causality and confounding remains problematic.

Prospective population studies on vitamin D (25(OH)D)) and lung function decline or incident COPD have yielded inconsistent results, with some reporting vitamin D deficiency or low vitamin D (25(OH)D) to be associated with a greater decline in lung function or increased risk of COPD compared with those with sufficient levels,[10 15 20] while others report null findings.[11 16] Small-scale genetic studies on polymorphisms in the vitamin D binding protein and rate of decline in lung function are also inconclusive.[9 40] Although we did not have incident COPD, we have shown that vitamin D deficiency (25OHD<10 ng/mL) was related to COPD mortality in all men except those with severe COPD; this was not explained by potential confounders and persisted after taking IL-6 into account. Our findings that COPD mortality in those with mild/moderate COPD is significantly increased only in those with vitamin D deficiency is consistent with findings from a randomised control trial (RCT) in subjects with COPD in which a post-hoc analysis was suggestive of reductions in exacerbations in the vitamin D supplementation group when subjects had what was regarded as severe vitamin D deficiency (25OHD<10 ng/mL) only.[41]

It is possible that vitamin D does not play a direct role in the progression of COPD but may prevent the development of COPD and COPD deaths through its prevention of respiratory tract infections. Evidence from both

observational and RCTs have shown that vitamin D supplementation prevents respiratory tract infections[42] which are a frequent complication of COPD and have been implicated in the onset and progression and exacerbation of chronic lung disease.[43] The lack of association with COPD deaths in those with severe COPD may be due to the advancement of the disease which may not be viable to detect any benefits of vitamin D. Alternately, this may be due to lack of power as 25(OH)D levels in these patients are low and relatively few patients have sufficient 25(OH)D levels.

## Vitamin D and total mortality in COPD subjects

Relatively few studies have examined 25(OH)D (vitamin D) and mortality in subjects with COPD and of those that have, many have reported null results and most have targeted moderate to very severe COPD patients.[20 21 23 24] Two of the population-based studies that have examined the association between 25(OH)D and mortality stratified by lung function status did not show low 25(OH)D (variously defined) to be independently associated with increased mortality.[21 22] However, in both these studies those with severe COPD were included and vitamin D deficiency (25(OH)D<10 ng/mL) specifically was not examined. Thus, the null findings with mortality in previous reports may be due to either the inclusion of those with severe COPD who are at an advanced stage of the disease, in whom vitamin D treatment will have little effect and/or the levels of 25(OH)D studied. However, two recent prospective studies have shown that 25(OH)D or low 25(OH)D (defined as <12.5 nmol/L) was associated with all-cause mortality in those with COPD in the general population.[25 26] A recent Cochrane review of vitamin D supplementation trials provided evidence that vitamin $D_3$ supplementation seem to decrease mortality in elderly adults.[44]

## Strengths and limitations

This study is based on a cohort of older (60–79-year-old) men who constitute a high-risk group for vitamin D deficiency and COPD. The study population is socially representative of the UK, and follow-up rates in the British Regional Heart Study are exceptionally high. We were able to take into account a wide range of confounders and inflammatory markers. Our biochemical assays were based on routine clinical assays and are therefore robust. However, blood measurements were based on a single measurement, raising the possibility that the strengths of associations may have been underestimated. In addition, we cannot preclude the possibility of residual confounding. It was based on an older predominantly white male population of European extraction, so that the results cannot be generalised directly to women, younger populations or other ethnic groups. Like most other epidemiological studies, we did not have lung function postbronchodilator. Although the present study was a prospective observational study and not a randomised trial, the association between vitamin D deficiency and

mortality appeared independent of confounding factors and inflammation.

## Conclusion

Men with COPD but not men with restrictive lung function were more likely to be vitamin D deficient than those with normal lung function. Vitamin D deficiency was associated with increased total mortality and mortality from respiratory and COPD causes in older men, and this was seen in those without lung impairment and in those with earlier stages of lung impairment. Intervention trials in older people with mild or moderate impaired lung function are needed to confirm whether increasing vitamin D levels through supplements in those with vitamin D deficiency will reduce both risk of COPD deaths and overall mortality.

**Contributors** SGW initiated the concept and design of the paper, analysed the data and drafted the manuscript. PWe and PWh contributed to the interpretation of data. OP contributed to the analysis of the paper. PWe, LL and PWh contributed to the acquisition of the data. All authors revised it critically for important intellectual content and approved the final version of the manuscript. SGW is the guarantor.

**Funding** The British Regional Heart Study is a Research Group supported by the British Heart Foundation (BHF) Programme grant (RG/19/4/34452).

**Competing interests** None declared.

**Patient consent for publication** Not required.

**Ethics approval** This study involves human participants and was approved by Ethics committee and ID. The National Research Ethics Service (NRES) Committee for London Ref MREC/02/2/91 ID191747. Participants gave informed consent to participate in the study before taking part.

**Provenance and peer review** Not commissioned; externally peer reviewed.

**Data availability statement** Data are available upon reasonable request. The data that support the findings of this study are available from the corresponding author upon reasonable request (contact Lucy Lennon at l.lennon@ucl.ac.uk)

**ORCID iD**
S Goya Wannamethee http://orcid.org/0000-0001-9484-9977

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
