## [Reviewer comments · BMJ Open]

ARTICLE DETAILS

TITLE (PROVISIONAL)	Vitamin D deficiency, impaired lung function and total and respiratory mortality in older men: The British Regional Heart Study
AUTHORS	Wannamethee, Goya; Welsh, Paul; Papacosta, Olia; Lennon, Lucy; Whincup, Peter

VERSION 1 – REVIEW

REVIEWER	rhodes, jonathan university of liverpool, school of clinical sciences
REVIEW RETURNED	24-May-2021

GENERAL COMMENTS	Good sized study, clearly presented. Some points for attention: 1. Statistical analysis - should state how you selected the variables included in your multivariate analysis.2. Adjustment for BMI - you have adjusted for BMI (I think - not mentioned in methods p 10 but is mentioned in footnotes to Tables 2-5). This is common practice but there is quite likely a causal association between high BMI and 25(OH)D deficiency - eg mediated by reduced hepatic 25-hydroxylase activity eg PMID 33210060 and PMID30790351 so adjustment for BMI is likely to reduce strength/size of any genuine associations with 25(OH)D deficiency. You could make this point in discussion and consider an additional supplementary analysis without adjustment for BMI.3. When looking at total mortality in relationship to 25(OH)D status it would be helpful to show data according to different categories of 25(OH)D sufficiency to address the controversial "U-shaped (or J-shaped) curve" that has been variably reported eg for example the large Danish general practice study by Durup et al PMID25710567 - this suggests increased all-cause mortality not only for deficiency but also for high 25(OH)D levels - it would therefore be helpful to see unadjusted/adjusted data for all-cause mortality according to further 25(OH)D concentration categories in the "sufficient" ranges eg 20-29, 30-39, ≥40 as well as <10 and 10-19 - this could be presented as a supplementary analysis. There is a plausible biological explanation for a possible harmful effect of high (non-toxic) 25(OH)D levels - mediated by induction of FGF23 and consequent inhibition of 1-hydroxylase eg Zitterman et al PMID32855522
--

REVIEWER	Holick, Michael Boston University School of Medicine, Vitamin D Research
-----------------	---

REVIEW RETURNED	27-May-2021
-------------

GENERAL COMMENTS	1. The authors used the term vitamin D levels. Since they did not measure vitamin D levels that should be corrected throughout the manuscript. Also the abbreviation for 25-hydroxyvitamin D is 25(OH)D 2. Since many medical societies have a different definition for vitamin D deficiency and insufficiency the authors should have at least acknowledge the difference. More importantly it would be of great interest to see whether there was any significant improvement in lung function for those who had circulating concentrations of 25 hydroxyvitamin D 30 ng/mL and greater for table 1. It would also be interesting to see how many had blood levels of at least 30 ng/mL. 3. The conclusion is weak in light of the results it would seem that it would be reasonable to suggest that vitamin D deficiency is associated with increased morbidity and mortality as the data suggests. 4. Was there any seasonal effect observed on either blood levels of 25 hydroxyvitamin D and/or lung function?
--

REVIEWER	Heath, Alicia Imperial College London, School of Public Health
-----------------	---

REVIEW RETURNED	08-Jun-2021
-------------

GENERAL COMMENTS	This study used data on 3575 men from the British Regional Heart Study to investigate the relationship between 25-hydroxyvitamin D concentrations and restrictive and obstructive lung function impairment, and assessed whether vitamin D deficiency was associated with mortality in those with impaired lung function. The results showed that participants with moderate and severe COPD were more likely to have vitamin D deficiency. Vitamin D deficiency was associated with a higher risk of total mortality and respiratory (and COPD) mortality in men with restrictive lung disease or mild/moderate COPD. Several sections of the manuscript need to be rephrased to improve clarity. It is important to distinguish between the cross-sectional analyses (in which vitamin D status and lung function/COPD were assessed at the same time point) and the prospective evaluation of 25OHD in relation to all-cause and respiratory or COPD mortality. Throughout the manuscript, what was assessed prospectively appears to have been confused. Some of the results are presented in a misleading way because the study did not evaluate the prospective association between 25OHD and incident risk of impaired lung function or COPD. Therefore, the data presented in tables 1 and 2, and the description of the analyses and results in these tables needs to be improved. Specific comments are detailed below. Page and line numbers refer to the numbers in the generated PDF (i.e. line numbers as written on the left side of each page).
---

	Page 1: affiliation number 4 for Peter H Whincup – is this meant to be labelled as 3? Page 2: Abstract, “HF” please define HF when first mentioned here. Definitions of vitamin D insufficiency and deficiency are controversial and there is no consensus on what is considered to be “normal”. It would be better to describe 25OHD>20 ng/ml as sufficiency or adequacy rather than normal. In the sentence “vitamin D deficiency was associated with moderate COPD “ what is described here appears to be the cross-sectional correlations presented in table 1. Since these were assessed at the baseline examination, perhaps this would be better described as “Men with moderate or severe COPD were more likely to have vitamin D deficiency...” Please see comment below about Table 1 – the %s are unclear. Page 5, line 12: “circulating 25-dehydroxyvitamin D”. This is incorrect. It should be 25-hydroxyvitamin D Page 5, Line 50: “vitamin D deficiency or low vitamin D” – these are essentially the same thing. “low” is non-specific. What is meant by low here? Was this meant to say “vitamin D deficiency or insufficiency...”? Page 5, Line 56: it does not make sense to say “circulating vitamin D deficiency”. The word circulating could be deleted, or this should be “circulating 25OHD” Page 7, first line: vitamin D is not the metabolite that is measured, so this should be “Blood measurements of 25OHD....” Page 7, Line 46: what do you mean by “total 25OHD2”? Is this meant to be “both 25OHD3 and total 25OHD” or “both 25OHD3 and 25OHD2”? Page 8, Line 29: Please define GOLD when first mentioned here – it is in the abstract but needs to also be spelled out when first mentioned in the main text. Page 8, Line 57: recommend elaborating on “tagging” or describe this in an alternative way for the benefit of readers who are not familiar with the UK NHS. Page 9, Line 17: There is not really a true definition of “normal” and it is meaningless to describe “normal vitamin D”. It might be better to say adequate or sufficient. Page 9, lines 24-27: Please give the categories of these variables, e.g. was it smoking status (never, former, current)? What were the categories of physical activity? How was alcohol intake categorised? Page 8, Line 28: typographical error “was fitted as fitted” – can delete as fitted Page 10, line 3: when assessing the exposure in relation to baseline characteristics, it is recommended that these are described qualitatively without attempting to report whether there
--	---

	are associations based on p-values. These shouldn't be described as "significantly" associated because significant could be interpreted to mean a large magnitude of effect, yet it is possible for p-values to be small (i.e. statistically "significant") despite moderate or small differences if a sample size is large enough. It is therefore recommended that the word significant is avoided here. Page 10, line 8: "lung function improved with increasing vitamin D levels" implies that this was a prospective association when this was a correlation. Do you mean men with better lung function were more likely to have higher 25OHD? Page 10, Line 10: this should be "increasing 25OHD concentrations" not "vitamin D levels" "deficiently" should be deficiency. Also, this sentence is unclear. Page 10, Line 24: if referring to vitamin D status, this should be "Circulating 25OHD was...." not "vitamin D was ..." Page 10, line 31: "odds"- how was this assessed? Is it the odds of having vitamin D deficiency in men with severe/moderate/mild COPD? This analysis was not described in the methods. It is unclear what the comparison is for the estimates presented in Table 2. Is it the odds ratio for having vitamin D deficiency versus sufficiency in each of the categories of lung function? This paragraph probably needs to be rephrased. Page 10, line 45: should be "25OHD concentration was inversely..." not "vitamin D was significantly and inversely....". Also, do you mean statistically significantly? The word significantly is not needed here. Perhaps give the HR and 95% CI at the end of this sentence. Page 10, line 52: it is important to specify what this is in comparison to, i.e. "for those with vitamin D deficiency and insufficiency, respectively, compared with vitamin D sufficiency." Page 10, line 57: after adjustment for what? Page 11, line 3 "significantly increased" – written this way, significant might be interpreted to mean a large magnitude instead of statistically significant Page 11, line 10: do you mean "statistically non-significant"? Page 11, Line 15 and Line 24: ""significant" – do you mean statistically significant? Page 11, line 40: should be circulating 25OHD not vitamin D Page 12, "we have shown vitamin D deficiency to be associated with increased risk of having COPD...". The study was not designed to be able to assess this. Vitamin D status and COPD were assessed at the same time point, so this was a cross-sectional evaluation of the prevalence of vitamin D deficiency in those with COPD, and all that can be inferred is that those with COPD were more likely to have vitamin D deficiency than sufficiency. Page 13, line 52: "vitamin D or low vitamin D" – is this referring to 25OHD?
--	---

	Page 14, Conclusion: this study did not evaluate vitamin D supplementation, so the conclusion detracts from what this study investigated. The conclusion should summarise the findings of this study without making a claim about supplementation (because supplementation was not assessed in this investigation). Page 15, “vitamin D levels” – should be 25OHD levels Table 1: the %s are unclear. Are they row or column %? These are difficult to interpret. Would it be clearer to report the % in each category of 25OHD, in which case the % should add to 100% going across the row? As mentioned in the comments above, p-values in a descriptive table should not be used to evaluate whether there is an association. A small p-value could simply be due to a large sample size when there might not be a clinically relevant difference (based on the magnitude of the difference). Table 1 and table 3 title: should be 25-hydroxyvitamin D not “vitamin D” levels Table 2 title: should be serum 25-hydroxyvitamin D not vitamin D As per previous comments, the analyses presented in this table need to be described in the methods. Table 3 header row: should be 25OHD not vitamin D
--	--

VERSION 1 – AUTHOR RESPONSE

Reviewer: 1

Prof. Jonathan Rhodes, university of Liverpool

Comments to the Author:

Good sized study, clearly presented. Some points for attention:

1. Statistical analysis - should state how you selected the variables included in your multivariate analysis.

Response: Thank you. We have now provided this information in Statistical Methods (page 9) “All analyses were initially adjusted for age and season. In the multivariate analyses we adjusted further for potential confounders known to be associated with lung function and mortality which included smoking, social class, physical activity, heavy drinking, BMI use of antihypertensive treatment, diabetes and pre-existing CVD. We also carried out supplementary analysis without inclusion of BMI since BMI may be a confounder or a potential mediator. We further adjusted for IL-6 as a potential mediator”.

2. Adjustment for BMI - you have adjusted for BMI (I think - not mentioned in methods p 10 but is mentioned in footnotes to Tables 2-5). This is common practice but there is quite likely a causal association between high BMI and 25(OH)D deficiency - eg mediated by reduced hepatic 25-hydroxylase activity eg PMID 33210060 and PMID30790351 so adjustment for BMI is likely to reduce strength/size of any genuine associations with 25(OH)D deficiency. You could make this point in discussion and consider an additional supplementary analysis without adjustment for BMI.

Response: Excluding BMI from the model made minor difference to the results. We have now provided models including and excluding BMI in the adjustment in Tables 3 and 4 (New model 2 and3)

3. When looking at total mortality in relationship to 25(OH)D status it would be helpful to show data according to different categories of 25(OH)D sufficiency to address the controversial "U-shaped (or J-shaped) curve" that has been variably reported eg for example the large Danish general practice study by Durup et al PMID25710567 - this suggests increased all-cause mortality not only for deficiency but also for high 25(OH)D levels - it would therefore be helpful to see unadjusted/adjusted data for all-cause mortality according to further 25(OH)D concentration categories in the "sufficient" ranges eg 20-29, 30-39, ≥ 40 as well as < 10 and 10-19 - this could be presented as a supplementary analysis. There is a plausible biological explanation for a possible harmful effect of high (non-toxic) 25(OH)D levels - mediated by induction of FGF23 and consequent inhibition of 1-hydroxylase eg Zitterman et al PMID32855522

Response: We have now provided this data in a Supplementary Table (S1) with those with sufficient levels further divided as suggested into 20-29, 30-39 and > 40 ng/ml. The number of men with levels > 40 ng/ml was small but there was no evidence that mortality was increased in these men. We have made additional comments in Discussion (page 13) and cited the reference by Durup et al (new ref 39)

Reviewer: 2

Dr. Michael Holick, Boston University School of Medicine

Comments to the Author:

1. The authors used the term vitamin D levels. Since they did not measure vitamin D levels that should be corrected throughout the manuscript. Also the abbreviation for 25-hydroxyvitamin D is 25(OH)D.

Response: Thank you. We have now changed "vitamin D levels" to 25(OH)D throughout the manuscript. The abbreviation 25(OH)D is now used throughout the manuscript.

2. Since many medical societies have a different definition for vitamin D deficiency and insufficiency the authors should have at least acknowledge the difference.

Response: Thank you. We have now added comments on the different definitions of vitamin D deficiency and sufficiency in Discussion Pages 12-13. We have added 3 new references relating to this (new reference 36-38). In particular we refer to the different definitions used by the NIH (US) and the NICE guidelines in the UK.

3. More importantly it would be of great interest to see whether there was any significant improvement in lung function for those who had circulating concentrations of 25 hydroxyvitamin D 30 ng/mL and greater for table 1. It would also be interesting to see how many had blood levels of at least 30 ng/mL.

Response: We have now provided this additional data stratifying the "sufficient group (> 20 ng/ml)" into sufficient ranges 20-29, 30-39 and > 40 in a Supplementary Table (S1). There was no evidence that lung function improved with increasing vitamin D levels in this "sufficient range". We have added comments in results describing the findings (page 10).

4. The conclusion is weak in light of the results it would seem that it would be reasonable to suggest that vitamin D deficiency is associated with increased morbidity and mortality as the data suggests.

Response: We have now modified the conclusion in Abstract and Discussion. Abstract conclusion Men with COPD were more likely to be vitamin D deficient than those with normal lung function. Vitamin D deficiency is associated with increased all-cause mortality in older adults with no lung impairment as well as in those with restrictive or obstructive lung impairment.

5. Was there any seasonal effect observed on either blood levels of 25 hydroxyvitamin D and/or lung function?

Response: There were differences in vitamin D and prevalence of COPD between those measured in winter and summer months. We have added this information in Results (page 10). However, season had been included in all the adjustments in Tables 2-5.

Reviewer: 3

Dr. Alicia Heath, Imperial College London

Comments to the Author:

1. Several sections of the manuscript need to be rephrased to improve clarity. It is important to distinguish between the cross-sectional analyses (in which vitamin D status and lung function/COPD were assessed at the same time point) and the prospective evaluation of 25OHD in relation to all-cause and respiratory or COPD mortality. Throughout the manuscript, what was assessed prospectively appears to have been confused. Some of the results are presented in a misleading way because the study did not evaluate the prospective association between 25OHD and incident risk of impaired lung function or COPD. Therefore, the data presented in tables 1 and 2, and the description of the analyses and results in these tables needs to be improved.

Response: We have now distinguished cross sectional analyses from the prospective analyses and made it clear that Tables 1 and 2 were cross sectional and Table 2-5 were prospective. We have altered the wording throughout to remove confusion. We have emphasised in discussion that the findings on vitamin D and lung function is cross-sectional.

We have added "cross-sectional" where relevant and highlighted this throughout the manuscript. For example, the new heading reads "Cross sectional associations between restrictive and obstructive lung function patterns and 25(OH)D and vitamin D deficiency"

2. Page 1: affiliation number 4 for Peter H Whincup – is this meant to be labelled as 3?

Response: Yes thank you.

3. Page 2: Abstract, "HF" please define HF when first mentioned here.

Response: Done

4. Definitions of vitamin D insufficiency and deficiency are controversial and there is no consensus on what is considered to be "normal". It would be better to describe 25OHD > 20 ng/ml as sufficiency or adequacy rather than normal.

Response: Thank you we have now changed this to sufficiency throughout. We have added in discussion some comments on the different definitions by the different societies (pages 13-14).

5. In the sentence "vitamin D deficiency was associated with moderate COPD" what is described here appears to be the cross-sectional correlations presented in table 1. Since these were assessed at the baseline examination, perhaps this would be better described as "Men with moderate or severe COPD were more likely to have vitamin D deficiency..."

Response: Thank you we have now modified the wording as suggested. We have changed the wording "associated with" in the cross sectional analyses and emphasised prevalence.

For example in Table 1 we now state "Men with vitamin D deficiency had the highest prevalence of COPD and this was seen for moderate and severe COPD but not for mild COPD."

In Table 2 we now state "Men with moderate or severe COPD were more likely to have vitamin D deficiency..."

7. Page 5, line 12: "circulating 25-dehydroxyvitamin D". This is incorrect. It should be 25-hydroxyvitamin D Page 5, Line 50: "vitamin D deficiency or low vitamin D" – these are essentially the same thing. "low" is non-specific. What is meant by low here? Was this meant to say "vitamin D deficiency or insufficiency..."?

Page 5, Line 56: it does not make sense to say "circulating vitamin D deficiency". The word circulating could be deleted, or this should be "circulating 25OHD"

Page 7, first line: vitamin D is not the metabolite that is measured, so this should be "Blood measurements of 25OHD..."

Response: This has been corrected.

8. Page 7, Line 46: what do you mean by “total 25OHD₂”? Is this meant to be “both 25OHD₃ and total 25OHD” or “both 25OHD₃ and 25OHD₂”?

Response: We apologise for the confusion. Total 25OHD includes (25OHD₂+25OHD₃). This has been corrected.

9. Page 8, Line 29: Please define GOLD when first mentioned here – it is in the abstract but needs to also be spelled out when first mentioned in the main text.

Response: We have defined GOLD. The Global Initiative for Chronic Obstructive Lung Disease

10. Page 8, Line 57: recommend elaborating on “tagging” or describe this in an alternative way for the benefit of readers who are not familiar with the UK NHS.

Response: We have now modified the text. Details on causes of deaths were collected through the National Health Service Central Register

11. Page 9, Line 17: There is not really a true definition of “normal” and it is meaningless to describe “normal vitamin D”. It might be better to say adequate or sufficient.

Response: We have changed this to “sufficient”.

12. Page 9, lines 24-27: Please give the categories of these variables, e.g. was it smoking status (never, former, current)? What were the categories of physical activity? How was alcohol intake categorised?

Response: We have now provided this information in methods in the section “Cardiovascular risk factor measurements at 1998-2000” (page 7)

13. Page 8, Line 28: typographical error “was fitted as fitted” – can delete as fitted Page 10, line 3: when assessing the exposure in relation to baseline characteristics, it is recommended that these are described qualitatively without attempting to report whether there are associations based on p-values. These shouldn’t be described as “significantly” associated because significant could be interpreted to mean a large magnitude of effect, yet it is possible for p-values to be small (i.e. statistically “significant”) despite moderate or small differences if a sample size is large enough. It is therefore recommended that the word significant is avoided here.

Response: We have omitted the word significant as suggested.

14. Page 10, line 8: “lung function improved with increasing vitamin D levels” implies that this was a prospective association when this was a correlation. Do you mean men with better lung function were more likely to have higher 25OHD?

Response: Yes. We have now made this clear in the text.

15. Page 10, Line 10: this should be “increasing 25OHD concentrations” not “vitamin D levels” “deficiently” should be deficiency. Also, this sentence is unclear.

Page 10, Line 24: if referring to vitamin D status, this should be “Circulating 25OHD was....” not “vitamin D was ...”

Response: We have now modified the text

16. Page 10, line 31: “odds”- how was this assessed? Is it the odds of having vitamin D deficiency in men with severe/moderate/mild COPD? This analysis was not described in the methods. It is unclear what the comparison is for the estimates presented in Table 2. Is it the odds ratio for having vitamin D deficiency versus sufficiency in each of the categories of lung function? This paragraph probably needs to be rephrased.

Response: We have now added in Statistical Methods “Logistic regression was used to assess the relative odds of having vitamin D deficiency (yes/no) for the lung impairment groups compared to those with normal lung function”.

17. Page 10, line 45: should be “25OHD concentration was inversely...” not “vitamin D was significantly and inversely...”. Also, do you mean statistically significantly? The word significantly is not needed here. Perhaps give the HR and 95% CI at the end of this sentence.

Page 10, line 52: it is important to specify what this is in comparison to, i.e. “for those with vitamin D deficiency and insufficiency, respectively, compared with vitamin D sufficiency.”

Page 10, line 57: after adjustment for what?

Page 11, line 3 “significantly increased” – written this way, significant might be interpreted to mean a large magnitude instead of statistically significant Page 11, line 10: do you mean “statistically non-significant”?

Page 11, Line 15 and Line 24: “significant” – do you mean statistically significant?

Response pages 10-11. Thank you. We have clarified all these issues regarding significance. We have avoided the word significant except when referring specifically to statistical significance or non-statistical significance in which case we have added the word “statistical”. We have now clarified the reference group for comparisons throughout.

18. Page 11, line 40: should be circulating 25OHD not vitamin D Page 12, “we have shown vitamin D deficiency to be associated with increased risk of having COPD...”. The study was not designed to be able to assess this. Vitamin D status and COPD were assessed at the same time point, so this was a cross-sectional evaluation of the prevalence of vitamin D deficiency in those with COPD, and all that can be inferred is that those with COPD were more likely to have vitamin D deficiency than sufficiency.

Response: We have modified the wording “associated with” throughout when inferring cross sectional associations and made it very clear that we are referring to cross sectional associations see response to 5.

19. Page 13, line 52: “vitamin D or low vitamin D” – is this referring to 25OHD?

Response: yes.

20. Page 14, Conclusion: this study did not evaluate vitamin D supplementation, so the conclusion detracts from what this study investigated. The conclusion should summarise the findings of this study without making a claim about supplementation (because supplementation was not assessed in this investigation).

Page 15, “vitamin D levels” – should be 25OHD levels

Response: We have now modified the conclusion which now summarises the main findings. See Response to Reviewer 1 (point 4). Reference to suggestions about supplementation from our data has been deleted from the conclusion.

21. Table 1: the %s are unclear. Are they row or column %? These are difficult to interpret. Would it be clearer to report the % in each category of 25OHD, in which case the % should add to 100% going across the row?

As mentioned in the comments above, p-values in a descriptive table should not be used to evaluate whether there is an association. A small p-value could simply be due to a large sample size when there might not be a clinically relevant difference (based on the magnitude of the difference).

Response: We have clarified that Table 1 summarises baseline characteristics by the three 25(OH)D groups. Table 1 is showing the characteristics of those who are vitamin D deficient compared to the other 25(OH)D groups. We have made it clear that the % in Table 1 is within each category of 25(OH)D.

For example, 25.3% of those with vitamin D deficiency were smokers compared with only 9.5% in those who had vitamin D sufficiency. We have added in footnotes that % refers to the % of men with the characteristics within the 25(OH)D groups

22. Table 1 and table 3 title: should be 25-hydroxyvitamin D not “vitamin D” levels Table 2 title: should be serum 25-hydroxyvitamin D not vitamin D As per previous comments, the analyses presented in this table need to be described in the methods.
Table 3 header row: should be 25OHD not vitamin D

Response: Thank you. We have changed this to 25(OH)D throughout. We have added the methods used in Table 2 in the statistical methods. Logistic regression was used to obtain relative odds of having vitamin D deficiency. ANOVA was used to obtain adjusted mean differences.

VERSION 2 – REVIEW

REVIEWER	rhodes, jonathan university of liverpool, school of clinical sciences
REVIEW RETURNED	27-Jul-2021

GENERAL COMMENTS	All points have been satisfactorily addressed but there looks to be a small error in footnote to Table 3 - says "Model 1 adjusted for ageBMIetc. Model 2=Model 1 + BMI" - presumably model 1 was not adjusted for BMI?
--

REVIEWER	Holick, Michael Boston University School of Medicine, Vitamin D Research
REVIEW RETURNED	19-Aug-2021

GENERAL COMMENTS	Accepted
----------

REVIEWER	Heath, Alicia Imperial College London, School of Public Health
REVIEW RETURNED	01-Aug-2021

GENERAL COMMENTS	Thank you for addressing all reviewer comments satisfactorily. However, some suggested edits and corrections were ignored and have not been corrected. There are also a few additional areas that require revision. Page numbers below refer to the page numbers in the author manuscript document. Page 5: “Vitamin D deficiency (ascertained from measuring the circulating 25-dehydroxyvitamin D” As mentioned in the first review, “25-dehydroxyvitamin D” is incorrect. It is “25-hydroxyvitamin D” Page 5. 2nd last line: “circulating vitamin D deficiency” As mentioned in the first review, this should either be “circulating 25(OH)D” or “vitamin D deficiency” Page 6, aim 1: “investigate the association between vitamin D deficiency and both restrictive and obstructive lung function impairment”
--

	This implies that it was investigated prospectively. Although the revised version states in the methods and results that this is a cross-sectional analysis, it would be clearer if the way this aim is phrased is modified as well. Page 9: typo “were fitted as fitted as” hasn’t been corrected. Additional comments: Abstract: “vitamin D deficiency was more prevalent in those with moderate COPD.....” this sentence does not state what the reference group is – more prevalent in those groups compared to who? Compared to those with normal lung function/no lung impairment? Abstract conclusion: “in older adults” should be ‘in older men’ since this study only included men. Page 8: which group does FEV1/FVC = 0.70 belong to? It is not in the moderate or mild airflow obstruction groups (FEV1/FVC <0.70) but not in the restricted or normal groups either (FEV1/FVC >0.70). One of these should be either ≤ 70 or ≥ 70. Same issue for FVC=80%. Should normal be FVC\geq80% instead of FVC>80%? Page 9: “Cox's proportional hazards model was used to assess the multivariate-adjusted hazards ratio (relative risk)” – for what outcome? Need to say this is for mortality. Page 10, results: “mean 25(OH)D was higher in men examined in winter (Dec-February) than in those examined in summer (June-- August) 22.2 (9.4) vs 17.1 (7.4) ng/ml” – recommend checking this since 25(OH)D is usually higher in summer than winter. “The prevalence of COPD was also much higher in winter than in summer (17% vs 30.2%)” - are the %s the correct way around? Page 11: “Prospective associations between 25(OHD)2 and Total mortality” – “25(OHD)2” should be “25(OH)D” Page 13 “Levels of 50 nmol/L...”. It would be useful to specify “25(OH)D levels of 50 nmol/L,,,” Page 14: “Prospective population studies on vitamin D [25(OH)D] and lung function decline or incident COPD have yielded inconsistent results, with some reporting positive findings” – positive meaning what direction here? Is this for vitamin D deficiency, i.e. vitamin D deficiency was associated with lung function decline and incident COPD? Page 14: “in the vitamin D supplementation when subjects” – is it meant to be vitamin D supplementation group? Page 16, conclusion: “mortality from respiratory and COPD causes in older adults” – older adults should be older men since this study only evaluated men. Table 3: HR per 1 ng/ml increase in 25(OH)D is missing for CVD mortality Model 1.
--	--

	Also “25OHD” in table header row and “increase in 25OHD” should be written as “25(OH)D”
--	---

VERSION 2 – AUTHOR RESPONSE

Reviewer 1

Comments to the Author:

All points have been satisfactorily addressed but there looks to be a small error in footnote to Table 3 - says "Model 1 adjusted for ageBMIetc. Model 2=Model 1 + BMI" - presumably model 1 was not adjusted for BMI?

Response: Thank you. This error has now been corrected.

Reviewer: 3

Dr. Alicia Heath, Imperial College London

Comments to the Author:

Thank you for addressing all reviewer comments satisfactorily. However, some suggested edits and corrections were ignored and have not been corrected. There are also a few additional areas that require revision. Page numbers below refer to the page numbers in the author manuscript document.

Response: We apologise for having overlooked some of the suggested edits in the first review and thank Dr Heath for pointing out the errors in the manuscript. These have all been corrected. All edits have been carried out.

1 . Page 5: As mentioned in the first review, “25-dehydroxyvitamin D” is incorrect. It is “25-hydroxyvitamin D”

Page 5. 2nd last line: “circulating vitamin D deficiency”. This should either be “circulating 25(OH)D” or “vitamin D deficiency”

Response: “dehydro. “ has been corrected . Circulating vitamin D deficiency has been changed to vitamin D deficiency.

2. Page 6, aim 1: “investigate the association between vitamin D deficiency and both restrictive and obstructive lung function impairment”

This implies that it was investigated prospectively. Although the revised version states in the methods and results that this is a cross-sectional analysis, it would be clearer if the way this aim is phrased is modified as well.

Response: we have added the word “cross sectional”. The sentence now reads “investigate the cross-sectional association between vitamin D deficiency and both restrictive and obstructive lung function impairment”.

3. Page 9: typo “were fitted as fitted as” hasn’t been corrected.

Response Now corrected.

Additional comments:

4. Abstract: “vitamin D deficiency was more prevalent in those with moderate COPD.....” this sentence does not state what the reference group is – more prevalent in those groups compared to who? Compared to those with normal lung function/no lung impairment?

Abstract conclusion: “in older adults” should be “in older men” since this study only included men.

Response: we have now added “compared to the normal lung function group”. Older adults have been changed to older men.

5. Page 8: which group does FEV1/FVC = 0.70 belong to?
Same issue for FVC=80%. Should normal be FVC≥80% instead of FVC>80%?

Response: Thank you this has been changed to >

6. Page 9: "Cox's proportional hazards model was used to assess the multivariate-adjusted hazards ratio (relative risk)" – for what outcome? Need to say this is for mortality.

Response: we have added the term mortality

7. Page 10, results:

"mean 25(OH)D was higher in men examined in winter (Dec-February) than in those examined in summer (June-- August) 22.2 (9.4) vs 17.1 (7.4) ng/ml" – recommend checking this since 25(OH)D is usually higher in summer than winter.

"The prevalence of COPD was also much higher in winter than in summer (17% vs 30.2%)" - are the %s the correct way around?

Response: Thank you for pointing out these errors. These have now been reversed. We have changed the wording to " ..mean 25(OH)D is lower in men examined in winter..." The prevalence in brackets is 30.2% vs 17%.

8. Page 11: "Prospective associations between 25(OH)D² and Total mortality" – "25(OH)D²" should be "25(OH)D"

Page 13 It would be useful to specify "25(OH)D levels of 50 nmol/L,"

Response: Thank you. These edits have been done.

9. Page 14: "Prospective population studies on vitamin D [25(OH)D] and lung function decline or incident COPD have yielded inconsistent results, with some reporting positive findings" – positive meaning what direction here? Is this for vitamin D deficiency, i.e. vitamin D deficiency was associated with lung function decline and incident COPD?

Response. This sentence has been clarified and now reads "...some reporting vitamin D deficiency or low vitamin D [25(OH)D] to be associated with greater decline in lung function or increased risk of COPD compared to those with sufficient levels".

10. Page 14: "in the vitamin D supplementation when subjects" – is it meant to be vitamin D supplementation group?

Page 16, conclusion: " older adults should be older men since this study only evaluated men.

Response: "group" has been added. Adults have been changed to "men".

11. Table 3: HR per 1 ng/ml increase in 25(OH)D is missing for CVD mortality Model 1.
Also "25OHD" in table header row and "increase in 25OHD" should be written as "25(OH)D"

Response: Thank you. This has now been added and header corrected.